# Chitosan-Human Bone Composite Granulates for Guided Bone Regeneration

**DOI:** 10.3390/ijms22052324

**Published:** 2021-02-26

**Authors:** Piotr Kowalczyk, Rafał Podgórski, Michał Wojasiński, Grzegorz Gut, Witold Bojar, Tomasz Ciach

**Affiliations:** 1Department of Biotechnology and Bioprocess Engineering, Faculty of Chemical and Process Engineering, Warsaw University of Technology, Waryńskiego 1, 00-645 Warsaw, Poland; rafal.podgorski.dokt@pw.edu.pl (R.P.); michal.wojasinski@pw.edu.pl (M.W.); tomasz.ciach@pw.edu.pl (T.C.); 2Warsaw University of Technology, CEZAMAT, Poleczki 19, 02-822 Warsaw, Poland; 3Department of Transplantology and Central Tissue Bank, Medical University of Warsaw, Chałubińskiego 5, 02-004 Warsaw, Poland; grzegorz.gut@wum.edu.pl; 4Dental Practice Witold Bojar, Opaczewska 43/124B, 02-201 Warsaw, Poland; witoldbojar@witoldbojar.pl

**Keywords:** guided bone regeneration, chitosan, allografts, cell scaffolds, thermal sterilization

## Abstract

The search for the perfect bone graft material is an important topic in material science and medicine. Despite human bone being the ideal material, due to its composition, morphology, and familiarity with cells, autografts are widely considered demanding and cause additional stress to the patient because of bone harvesting. However, human bone from tissue banks can be used to prepare materials in eligible form for transplantation. Without proteins and fats, the bone becomes a non-immunogenic matrix for human cells to repopulate in the place of implantation. To repair bone losses, the granulate form of the material is easy to apply and forms an interconnected porous structure. A granulate composed of β-tricalcium phosphate, pulverized human bone, and chitosan—a potent biopolymer applied in tissue engineering, regenerative medicine, and biotechnology—has been developed. A commercial encapsulator was used to obtain granulate, using chitosan gelation upon pH increase. The granulate has been proven in vitro to be non-cytotoxic, suitable for MG63 cell growth on its surface, and increasing alkaline phosphatase activity, an important biological marker of bone tissue growth. Moreover, the granulate is suitable for thermal sterilization without losing its form—increasing its convenience for application in surgery for guided bone regeneration in case of minor or non-load bearing voids in bone tissue.

## 1. Introduction

Bone grafts are considered most often transplanted tissue after blood, with more than half a million procedures just in the United States [1] being necessary in cases of bone resections. Numerous surgical procedures require harvesting the patient’s own bone, which often causes discomfort and regeneration issues for the patient. Bone regeneration remains a challenge for tissue engineering where several factors should be considered when designing the solution: Providing structural support in the place of missing bone, enabling patient’s cells to grow and regenerate the bone tissue; providing adequate substances to build the tissue from; biocompatibility and harmless absorption or integration of implanted materials. The “golden standard” in bone regeneration is grafting the patient’s bone, e.g., harvested from the iliac crest [2]. Such material meets osteoconduction, osteoinduction, and osteogenicity requirements, but does not always provide structural support. Besides, autografts’ harvesting procedure is highly uncomfortable and at the risk of complications [3]. Instead of using autografts, a popular procedure applies to human bone that has not originated in the patient’s body, called an allograft. Thanks to the potential presence of signal proteins, extracellular matrix, and mineral composition, allografts offer increased regeneration, a native environment for osteoblasts, and additional surgery elimination. Nevertheless, the principal risks of using those materials are potential host incompatibilities, contaminations, and disease transmissions [4,5,6]. An alternative for genuine bone grafts is synthetic materials supporting the regeneration: Calcium triphosphates [7], brushite [8], calcium sulfate [9], hydroxyapatite [10], also polymeric scaffolds obtained as granules [11] or 3D-printed [12]. Those approaches can be combined with 3D prints providing structural support and granulates, with osteoconductive and osteogenic ceramics [13]. Synthetic supports do not require the risky procedure of autograft harvesting. They can be fine-tuned for optimal resorption time, mechanical properties, and presence of various additives, such as antibiotics or bone growth factors [14].

For the construction of tissue regeneration scaffolds and void fillers, several classes of materials can be mentioned. The main classification takes the material’s chemical nature into account—ceramics, metals, or polymers [15]. This paper focuses on polymer and ceramics applications of materials. Within the polymers class, materials are often divided into synthetic and natural. Special synthetic polymers for scaffold construction include polylactic acid, polycaprolactone, polyethylene glycol, and polyglycolic acid, as well as their copolymers [16]. Properties of those materials, such as molecular weight, mechanical properties, and functional groups, can be fine-tuned on a molecular level during synthesis. Bone tissue engineering applications include also various natural polymers, such as alginate [17], agarose [18], chitosan [19,20], collagen [21], fibrin [22], and hyaluronic acids [23]. Those materials offer non-toxicity and biodegradability, but they are often difficult to process into scaffolds and stimulate inflammatory reactions [24]. However, natural biopolymers are valuable in hydrogel applications and are widely used as wound dressings, bioactive scaffolds, and drug delivery systems [25]. They offer properties suitable for cell growth support, and such materials do not have to be purified from polymerization reagents. The release of reagents and their presence in synthetic polymers might cause cytotoxic effects [26]. One of the most frequently used natural polymers, chitosan, is renowned for its excellent biocompatibility, adsorption after implantation, and antibacterial properties [27]. Chitosan, which chemically is an aminoglycan, has amino groups incorporated into the polysaccharide backbone. It is insoluble in water and basic pH, but in acidic pH, those amino groups become protonated, and the polymer becomes soluble. Increasing pH, e.g., by addition of sodium hydroxide or ammonia, deprotonates amino groups and causes gelation of the chitosan [28].

Composites made of polymers and ceramics represent a strategy that combines polymers’ flexibility with porosity and regenerative potential of minerals similar to bone [10,16]. The inclusion of ceramic material like hydroxyapatite and calcium phosphate in chitosan hydrogel allows one to obtain a potent bone regeneration composite [29,30]. Regenerative properties of a hybrid composite increase even more with the usage of genuine bone powder—a bioceramic. Dumas et al. prepared an injectable composite cement with human and bovine bone powder suspended in polyurethane, which mechanical properties were similar to calcium phosphate bone cement, but regenerative properties in rats were superior [31]. Genuine human bone granules are already applied to fill bone gaps. The disadvantage of granules is that they tend to move in the place of implantation. By mixing those granules with chitosan, a biocompatible hydrogel coating is formed, which enhances agglomeration and prevents the graft’s unwanted dispersion. Additionally, the granulation process controls the products’ size better, hence pore size between granules. The pore size in a scaffold plays an essential role in cell growth: It has been proven that pore size of 300–325 μm allows the most efficient bone cell growth [32]. The interconnecting pore size in a granular scaffold can be easily regulated with particle sizing. Depending on the further modification of the granulates, such materials could find applications in the small or non-load bearing bone void filling (unmodified granules) [33] or as scaffolds for almost every type of bone defect filling (granules fused in the form of the scaffold) [20,34].

This paper presents a method of obtaining chitosan-bioceramic granulate (Figure 1), and we describe its properties and bone regeneration potential for small or non-load bearing voids. We have used both commercially available β-tricalcium phosphate and genuine human bone to determine the effect of the natural-origin material presence on cell growth. This developed composite chitosan-calcium phosphate-human bone material could augment guided bone regeneration with higher efficiency than fully synthetic granulates and could provide an alternative to purely ceramic bone filler, xenografts, etc. This material is also sterilizable with an autoclave, a cheaper and more commonly available sterilization method than ethylene oxide or gamma radiation methods.

## 2. Results and Discussion

Chitosan (CS) solution was ejected from the encapsulator’s nozzle into a stirred water solution of sodium hydroxide, which resulted in the rapid gelation of chitosan and the formation of spheres. Both calcium phosphate (CS β-TCP) and calcium phosphate/human bone (CS β-TCP/HB) composites were formed in the same manner. Different from presented in this paper, concentrations of polymer, acetic acid, and sodium hydroxide were tested, and different encapsulator settings, but results were unsatisfactory, as granules were falling apart or encapsulator jet was failing. Upon thermal drying at 50 °C, it has been observed that CS spheres lost most of their volume and became flakes (results not shown). Dehydration via freeze-drying was used as an alternative, and granulates retained a more spherical shape, but they have lost weight as well, which suggests that CS particles were hydrogel particles (Figure 2 and Figure 3C).

This behavior is confirmed in the literature, as gelation of chitosan upon pH increase is a phenomenon widely used for chitosan hydrogel fabrication [35,36]. Sterilized composites retained their spherical shapes. Sterilization of materials with an autoclave resulted in degradation and carbonization of chitosan flakes, while ceramic composite materials remained intact, retaining their color and shape. This tendency is visible in size analysis: Differences between the mean size and modes of size between raw and sterilized composites are insignificant (*p* < 0.05), while pure chitosan granules were significantly shrunk after sterilization (Figure 3). A sharp cut-off of diameters below 0.5 mm is caused by the digital image analysis tuned to particles larger than the size mentioned above to avoid counting image artifacts and debris as particles. The mean diameter of CS particle decreased from 0.975 ± 0.053 mm to 0.663 ± 0.032 mm (~47%), while CS β-TCP granules shrank from 1.264 ± 0.025 mm to 1.242 ± 0.047 mm (~2%) and CS β-TCP/HB granules shrank from 1.523 ± 0.075 mm to 1.517 ± 0.108 mm (<1%). Those results show that both β-TCP and human bone powder are good stabilizers for chitosan during thermal sterilization and allow us to prepare larger particles. Despite different mean sizes of CS β-TCP and CS β-TCP/HB, their diameters’ modes are comparable. This leads to the conclusion that bone usage results in less monodisperse material, which can be observed in size distributions. This could be caused by the irregular shape of bone tissue particles compared to much finer particles of β-TCP reinforcing the composite. Incubation of materials in PBS (Figure 3B,C) revealed a nonsignificant mass increase, except for non-sterilized CS β-TCP granules, where the increase was minimal. The swelling ratio of 200% is significantly smaller than other findings on composite chitosan-ceramic materials, like 20 times the dry mass in composite chitosan-gelatin-nanohydroxyapatite-polyaniline porous scaffolds [37], but since the formulation is different—granules versus porous sponge—it is expected to observe less absorbed water in a granular structure with high ceramic content.

SEM revealed that CS material possesses a folded, smooth surface, and upon magnification to 25,000×, a structure of interwoven polymer nanofibers can be observed (Figure 2). This is in agreement with SEM photography of chitosan published by Kumar and Santosh [38], who observed texture changes upon thermal sterilization. CS particles become flakes, and sharp, crystalline structures replace the nanofibers observed in non-sterile material. Single nanocrystal size has been roughly measured to be 440 ± 100 × 170 ± 40 nm. Surfaces of composite materials are coarse and grainy, with numerous visible particles of β-TCP and bone. High magnification reveals the chitosan nanofibrous structure enveloping the ceramic particles. Sterilization in autoclave slightly modifies the appearance of the composite surfaces. At 25,000× magnification, the sterilized chitosan matrix with added β-TCP and bone particles is visibly less crystallized than the one from the sample without those components, and the nanofibrous structure is preserved. β-TCP and bone particles seem phenomenon might be caused by thermal degradation and partial melting of the chitosan, which is known for decreasing the polymer’s molecular weight and changing the mechanical properties during heating in the 180–310 °C temperature range [39,40]. However, thermogravimetric, calorimetric, and FTIR studies of the chitosan degradation process by Corazzari et al. [41] indicate that the polymer’s chemical decomposition does not peak until exceeding 280–310 °C, and the mass loss in lower temperatures is caused by water desorption. Those temperatures are significantly higher than 121 °C in the autoclave, so chitosan’s chemical decomposition is highly unlikely in this study’s sterilization conditions. There are reports on the degradation of chitosan powders and hydrogels after thermal sterilization, becoming yellowish and brown when exposed to long sterilization times [39,42,43]. Chitosan, especially one with a high deacetylation ratio, has numerous primary amine groups able to be used in Schiff bonds [44]. As a long-chained polysaccharide with glucosamine units linked with β-1,4 glycoside bonds, it cannot react with amines via aldehyde groups, as only the terminal sugar units can mutarotate. A hypothetical decrease of molecular weight might increase the number of terminal glucosamine units in chitosan chains, although organic chemistry of chitosan predicts reactions between them as unlikely. Instead of chemical crosslinking between chains, a purely physical change is possible. High-pressure water vapor penetration, melting of chitosan (as mentioned earlier), and subsequent exposition to vacuum change the chain packing, and therefore, the polymer’s phase to be suspended in a smooth polymer matrix in the sterilized samples. This composition, thus creating the nanocrystalline structure visible in Figure 2.

Chitosan FTIR spectrum correlates well with other FTIR studies of chitosan. The spectra of sterilized and granulated chitosan do not differ, as has been studied by Corazzari et al., which has proven that the FTIR spectra of chitosan start changing upon thermal decomposition in temperatures much higher than in this work [41]. This result supports the previously presented hypothesis of chitosan not undergoing a chemical change upon sterilization. Analysis of ceramic materials revealed the presence of peaks at 400–1200 cm^−1^ in β-TCP and human bone powders and additional broad peaks at 1300–1600 cm^−1^ in the human bone sample (Figure 4). Those peaks are missing in pure chitosan samples, but are repeated in spectra of final materials, confirming ceramics’ presence in composites. Peaks at 561, 603, and 1020 cm^−1^ are characteristic for PO_3_^2−^ ions [45,46], and the human bone sample exhibits residual amide and CO_3_^2−^ peaks at 1300–1600 cm^−1^ typically found in bone tissue [47]. CS β-TCP/HB material FTIR spectrum shows absorption peaks originating from functional groups characteristic for ceramics and bone residuals, which confirms materials composition.

Performed observations of materials in high magnification SEM revealed a nanocrystal-coated surface of the sterilized chitosan. Those oblong nanocrystals appearing after thermal treatment are sized below 1 μm in length. A layer of crystalline chitosan creates a coarse surface, which could disrupt proper cell attachment, making adhesion and flattening difficult for the cell [48], and detaching nanocrystals could damage the cells via intake or mechanical puncture. Confocal microscopy of long-term culture of MG63 cells revealed that cells were poorly attached to pure chitosan materials, but they have entirely covered both chitosan-ceramic composites (Figure 5), which seems to support the hypothesis of unsuitable cell adhesion surface formation. Fluorescence of pure chitosan can be observed, which is consistent with literature describing fluorescence of chitosan at 400/470 nm excitation/emission wavelengths, which is comparable to 405/488 nm used for phalloidin imaging [49]. Cytotoxicity XTT assay confirmed that CS β-TCP/HB material is within the 70% L929 cells viability norm, but pure chitosan and CS β-TCP materials were more harmful to those cells in vitro (Figure 5). However, composite materials did not cause high cytotoxicity, consistent with other works where calcium phosphate/chitosan composites were obtained and sterilized in autoclave [50]. The chitosan’s toxicity was apparent when using extracts for viability assays, which suggests a mechanism of nanocrystals detaching from the material surface and damaging the cell culture in indirect contact.

The long-term culture of MG63 cells on composites was carried without visible toxic effects on cultured cells. Cells were cultured in 6-well plates, and the total protein concentration was constant for three weeks, except for control cells between weeks 1 and 2, where total protein significantly increased by 14%. Both CS β-TCP and CS β-TCP/HB composites induced an increase of ALP activity, with 21% higher activity in the bone-containing sample at day 21 compared to CS β-TCP and 27% compared to control (Figure 5). Control cells also exhibited similar behavior. The presented data shows that CS β-TCP/HB composite induces significantly higher ALP activity than CS β-TCP, which does not significantly differ from control after three weeks. Those results indicate that scaffolds enriched with pulverized bone and tricalcium phosphate promote osteogenic response significantly better than scaffolds loaded only with tricalcium phosphate, making them promising materials for bone tissue engineering. ALP is a commonly researched osteogenic marker, and its activity increases in 1–2 weeks of in vitro cultures, being one of the first enzymes participating in new bone formation. A universal increase in ALP expression after 1–2 weeks has been reported to be expected in the MG63 cell line [51,52], although other authors claim an increase in its activity happens faster after one week [53]. However, similar experiments performed on chitosan-hydroxyapatite-poly-3-hydroxybutyrate-co-3-hydroxyvalerate scaffolds revealed a similar ALP activity trend with an increase in both composite materials and control between 5 and 20 days of culturing [54]. ALP is considered a late osteogenic marker in hMSC cells, with a peak of activity after two weeks [55]. This unusual delay in the ALP activity in performed experiments could be caused by the time needed for colonization of chitosan composite scaffolds, with cells becoming enzymatically more active after attaching to granulates’ surfaces. This delay is visible in the confocal microscopy in Figure 5, where granulates are barely colonized after one week.

Chitosan, derived mainly from crustacean shells produced by the food industry, has been described before as a potent material in bone engineering, and is for its biochemical properties and because it is an environment-friendly marine material [56]. Compared to ceramic/polymer composite, bone-infused composites serve a specific purpose: To possess the base polymer-synthetic mineral composite’s mechanical properties, but to be more osteoconductive and osteoinductive. This strategy was used in other works aiming at bone-polymer composites and their comparison with ceramic-polymer composites [31]. The results of insignificant structural difference between β-TCP and human bone composites and increased cell viability on human bone-containing granulates are consistent with the results of more advanced materials using bone. Bone composite increases ALP activity in vitro compared to pure tricalcium phosphate, which is a promising result indicating potential properties leading to bone regeneration in filling small or non-load bearing voids in bone tissue [33]. However, to evaluate the regenerative properties of both granulates with and without human bone addition further, in vivo, tests on animals should be performed, which is the next step of research that is to be conducted. Both animal and in vitro studies should be considered limited models of human reactions, and the observed improvements should be carefully and systemically evaluated in the aspects of drug doses on animals, the timing of administration, and other interactions between the biomaterial, biological system, and the methodology of research, many of those being deceptive and unpredicted.

## 3. Materials and Methods

All used chemicals were analytical grade and used without further purification unless stated otherwise. Water was ultrapure grade: 40.1 µS∙cm^−1^ TC (purified with the Milli-Q system, Merck, Darmstadt, Germany).

1 g of chitosan (Chitoscience 95/2000, 95% deacetylation, medical-grade, Heppe Medical Chitosan GmbH, Saale, Germany) was dissolved in 100 g 1% (*w*/*v*) acetic acid (POCH S.A., Gliwice, Poland) in water solution under stirring for 4–6 h using a magnetic stirrer. According to standard procedures, a human cortical bone was procured, tested, processed, ground, and defatted by Training and Research Tissue Bank at National Centre for Tissue and Cell Banking in Warsaw, Poland. According to Polish transplantation law requirements, the Bank applies strict donor selection criteria, including biological tests, and processes tissues under controlled conditions. Degreased and the freeze-dried human cortical bone powder was milled in a Pulverisette 6 bead mill (Fritsch GmbH, Idar-Oberstein, Germany) for 2 h, 150 rpm to obtain ≤200 µm in size particles.

CS solution was then mixed with β-tricalcium phosphate (β-TCP, Sigma-Aldrich, St. Louis, USA) and human bone (HB) powders to obtain 10% (*w*/*w*) chitosan/β-tricalcium phosphate (CS β-TCP) suspension (5 g of β-TCP in 50 g of 1% chitosan solution) and 5%/5% (*w*/*w*) chitosan/tricalcium phosphate/human bone (CS β-TCP/HB) suspension (2.5 g of calcium phosphate and 2.5 g of milled bone in 50 g of 1% (*w*/*w*) chitosan solution). All solutions were used immediately after preparation and were not stored for later use to avoid acidic hydrolysis of the chitosan and ceramics degradation.

### 3.1. Preparation of Granulate

A B-395 Pro encapsulator (Büchi Labortechnik, Essen, Germany) was used to produce granulate from chitosan solution and ceramic suspensions. Liquids were fed into the encapsulator with a syringe pump at a 20 mL·min^−1^ feed rate. Encapsulator configuration was as follows: 450 µm nozzle, membrane vibration frequency 300 Hz, and stabilizing voltage 500 V. Chitosan suspensions were injected into a magnetically stirred coagulation bath 1% (*w*/*w*) sodium hydroxide (NaOH, POCH S.A., Gliwice, Poland) in water. Obtained spheres were stirred in NaOH solution for 1 h for proper gelation of the chitosan. Then spheres were washed with ethanol (POCH S.A) for 1 h and freeze-dried to obtain dry granulates. To investigate the effect of sterilization on the granule’s potential degradation, each type of material was sterilized for 20 min, 121 °C, in a AS 25 V steam autoclave (SMS Sp.z o.o., Góra Kalwaria, Poland). Materials were prepared using the most suitable parameters of the encapsulator for the production of stable granules.

### 3.2. Characterization of Granulate

Obtained granules were photographed against a black background; then, images were digitally processed with ImageJ software [57] to determine particles’ area. The size of particles was then calculated as sphere diameters, in which spheres had a cross-section area equal to the area of the measured particles. The only area of particles with circularity above 0.4 was processed to avoid counting small image artifacts as tiny particles and dominating the size distribution. Only particle counts with an area larger than 0.35 mm^2^ and circularity above 0.4 were processed. Experimental data is available in Appendix A.

Granules were imaged with a FESEM SU8230 scanning electron microscope (Hitachi, Tokyo, Japan). Before the SEM analysis, the samples were placed on conductive carbon/aluminum tape. An SEM stub with a sample was coated with a layer of about 10 nm of Au/Pd (80/20 mass ratio) using a Q150T sputter coater (Quorum Technologies, Laughton, UK) to enhance the conductivity of the samples. In the SEM system, a working distance of 6 mm was set up, and a 10 kV landing voltage was used to stream electrons to the surface of the sample. Secondary electron (SE) detectors (upper—U and lower—L) were used to collect the samples’ signal for low-magnification and high-magnification imaging. The presence of functional groups on the surfaces of three materials (chitosan, CS β-TCP, and CS β-TCP/HB) was investigated using Fourier-transform infrared spectroscopy (FTIR). A Nicolet 6700 spectrometer (Thermo Fisher Scientific, Waltham, USA) equipped with a SmartOrbit high-performance diamond single bounce ATR accessory was used. Spectra were analyzed using manufacturer-provided OMNIC 8.3 and processed using Prism 9 (GraphPad Software, San Diego, CA, USA) software. Experimental data is available in Appendix A.

Swelling studies were performed on all six variants of the material (chitosan, CS β-TCP, and CS β-TCP/HB, unsterilized and sterilized). Samples were dried by brief contact with filter paper and carefully weighed with a Balance XPR2 analytical weight (Mettler-Toledo, Greifensee, Switzerland) at 0, 24, and 48 h after placing in 0.1 M PBS containing 0.1% sodium azide (Sigma-Aldrich) for protection against microbial contamination, in 37 °C. Experimental data is available in Appendix A. Granulates before and after soaking were photographed with a Leica M205A stereoscopic microscope (Leica Camera AG, Wetzlar, Germany), using the Z-stacking image processing.

### 3.3. Materials Cytotoxicity and Cell Viability on Materials

The method of determining cytotoxicity of obtained granulates was based on ISO EN 10993-5 standard. Samples of materials were placed in open Eppendorf 5 mL tubes in self-sealing sterilization pouches and sterilized at 121 °C for 20 min. Next, 1 mg of each sample was put to wells on a 24-well plate. Then, 1 mL of Dulbecco’s Modified Eagle Medium with phenol red (DMEM, Thermo Fisher Scientific) supplemented with 10% (*v*/*v*) Fetal Bovine Serum (FBS, Thermo Fisher Scientific) and 1% (*v*/*v*) Penicillin-Streptomycin (Pen-Strep, 100 U∙mL^−1^ penicillin, 100 mg∙mL^−1^ streptomycin, both purchased from Thermo Fisher Scientific) solution as extraction medium was added to each well. Samples were incubated in the extraction medium for 24 h at 37 °C. A supplemented DMEM incubated in the same period was used as negative cytotoxicity control. L929 cell line (ATCC CCL-1, American Type Culture Collection, Manassas, USA) was seeded in 96-well plate in concentration 10^4^ cells per well in 100 µL of DMEM medium with phenol red supplemented with FBS 10% (*v*/*v*) and Pen-Strep 1% (*v*/*v*). After 24 h of incubation, the medium was removed from each well with cells, and 100 µL of samples from 24 h extraction in extraction medium and negative control, were added. After 24 h of incubation, extracts were removed, and each well was twice washed with Dulbecco’s Phosphate-Buffered Saline without calcium and magnesium ions (DPBS, Thermo Fisher Scientific). Then, 100 µL of DMEM without phenol red and supplementation and 50 µL of Cell Proliferation Kit II (XTT) (Hoffmann-La Roche, Basel, Switzerland) solution were added to each well. Cells with XTT in DMEM were incubated for 4 h. Then, 100 µL of cytotoxicity assay medium from each well was transferred to a new 96-plate, and the absorbance at 475 nm was measured in an Epoch spectrophotometer (BioTek, Winooski, USA). The relative cell viability was defined as the ratio of the mean absorbance of the sample to the mean absorbance of the negative control. Experimental data is available in Appendix A.The behavior of MG63 cells (human osteosarcoma cell line, Sigma-Aldrich) in a direct culture was determined by seeding 10^3^ cells on granulate samples and cultured in a 24-well plate with DMEM medium with phenol red supplemented with FBS 10% (*v*/*v*) and Pen-Strep 1% (*v*/*v*). Cells on granulate samples were cultivated for 1, 3, and 7 days with changing culture medium for a fresh one every two days. After each culture period, granulate samples were transferred to new 24-well plates and washed twice with DPBS without calcium and magnesium ions. Next, samples were soaked in 1 mL of 4% (*w*/*v*) water solution of paraformaldehyde (PFA, Sigma-Aldrich), 1 mL of 0.2% (*v*/*v*) water solution of Triton X-100 (Sigma-Aldrich), 200 µL of 165 nM of Alexa Fluor™ 488 Phalloidin (Thermo Fisher Scientific) solution in DPBS and 100 µL of 300 nM DAPI (Thermo Fisher Scientific) solution in DPBS; each step was separated by washing in 1 mL of DPBS twice. Samples were imaged with a LSM 880 confocal laser scanning microscope (Carl Zeiss AG, Jena, Germany).

### 3.4. Metabolic Activity and Mineralization

To determine the bone regeneration potential, alkaline phosphatase activity, a standard marker for bone regeneration, was determined in MG63 cells cultured on chitosan composites. MG63 cell line was cultured with the same medium and in the same conditions as for cytotoxicity assay, in 6-well plates, seeded with 10^4^ cells per well. Sterilized CH β-TCP and CH β-TCP/HB granulate were inserted directly into the medium in a quantity of 20 mg per well. The cell medium was changed every two days. After 7, 14, and 21 days of culture, the medium was removed, cells were washed three times with PBS, and 2 mL of cold RIPA buffer (Pierce Chemical, Dallas, USA) was added per well to disintegrate cells. After 10 min on a laboratory orbital shaker (ELMI Skyline, Riga, Latvia) on 400 RPM the fluid with granulate was transferred to plastic tubes and centrifuged at 14,400 RCF for 10 min (MPW-251 centrifuge manufactured by MPW Med. Instruments, Warsaw, Poland). Then, 300 μL of lysate was mixed with 150 μL of substrate buffer (10 mM p-nitrophenol phosphate, 5 mM MgCl_2_, 0.1 M diethanolamine in water, pH 9.5) and incubated at 37 °C. p-nitrophenol phosphate was purchased from Thermo Fisher Scientific, MgCl_2_ and diethanolamine were purchased from Sigma-Aldrich. After 2 h, the reaction was terminated with 300 μL of 1 M NaOH. The obtained yellow p-nitrophenol concentration was determined with a plate reader (BioTek) at 405 nm. The standard curve was prepared using dilutions of p-nitrophenol in ultrapure water alkalized to pH = 12 (R^2^ = 0.9993). The measured p-nitrophenol amount was expressed as nmol/min. The total protein amount was determined with the BCA assay (Quanti-Pro^TM^, Sigma-Aldrich). Bovine serum albumin standard (Sigma-Aldrich) was used for the standard curve (R^2^ = 0.998). Experimental data is available in Appendix A.

### 3.5. Statistical Analysis

Granulates were prepared in triplicates. Size distributions were prepared for collective particle counts, while mean sizes and modes of size were calculated for each repetition of granulates preparation, which compares the repeatability of the proposed process. Mean values of a measured size and calculated polydispersity index for variants of prepared materials were compared for the statistical difference using one-way ANOVA with post hoc Tukey test at *p* < 0.05. Before ANOVA, the normality of data points distribution was checked using the Shapiro–Wilk test at *p* < 0.05. Data analysis was performed using Origin 8 software (OriginLab Corporation, Northampton, USA).

## 4. Conclusions

In this study, a granulate-shaped composite of highly deacetylated chitosan, β-tricalcium phosphate, and pulverized human bone was obtained using encapsulation and granulation. The granules’ size can be tuned to obtain interconnecting pores of the desired size between the pellets, as pore size plays an essential role in vascularization and bone cell growth efficiency. Composite material retained its physicochemical properties after thermal sterilization, unlike pure chitosan granules. Cytotoxicity evaluation proved cell viability within the ISO norm. Confocal microscopy revealed stable MG63 cell growth on the surface of obtained materials—β-TCP/human bone composites performed significantly better than β-TCP composites, and induced increased alkaline phosphatase activity in cells. Obtained material can be easily prepared and thermally sterilized with an autoclave, a simple, low-priced, and commonly used procedure compared to expensive and specialized ethylene oxide or radiation methods. Further research, including in vivo tests, would be conducted to determine the prepared composite’s regenerative potential.

## Figures and Tables

**Figure 1 ijms-22-02324-f001:**
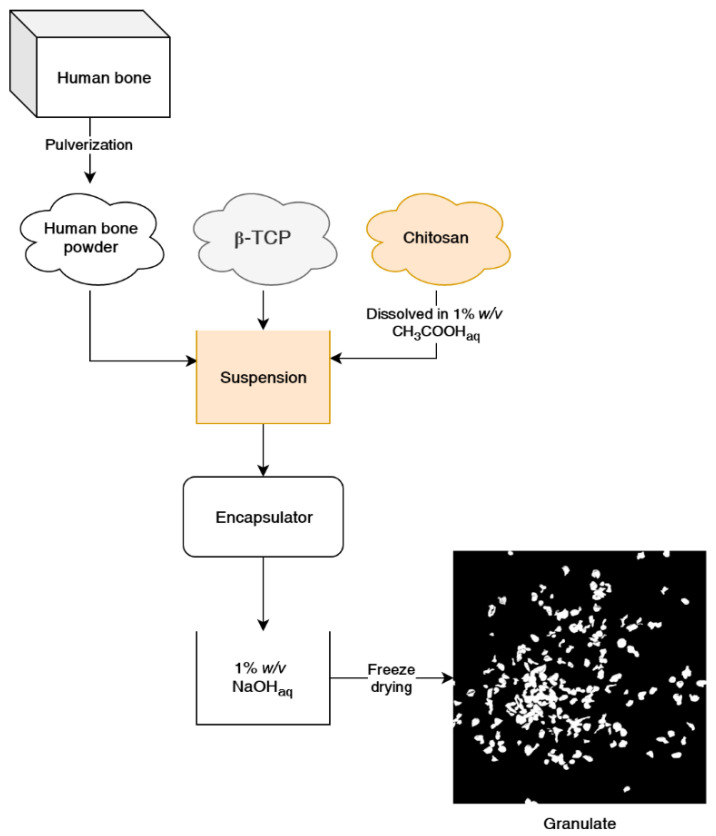
Schematic of the chitosan-β-tricalcium phosphate/human bone (CS β-TCP/HB) composite material formation process.

**Figure 2 ijms-22-02324-f002:**
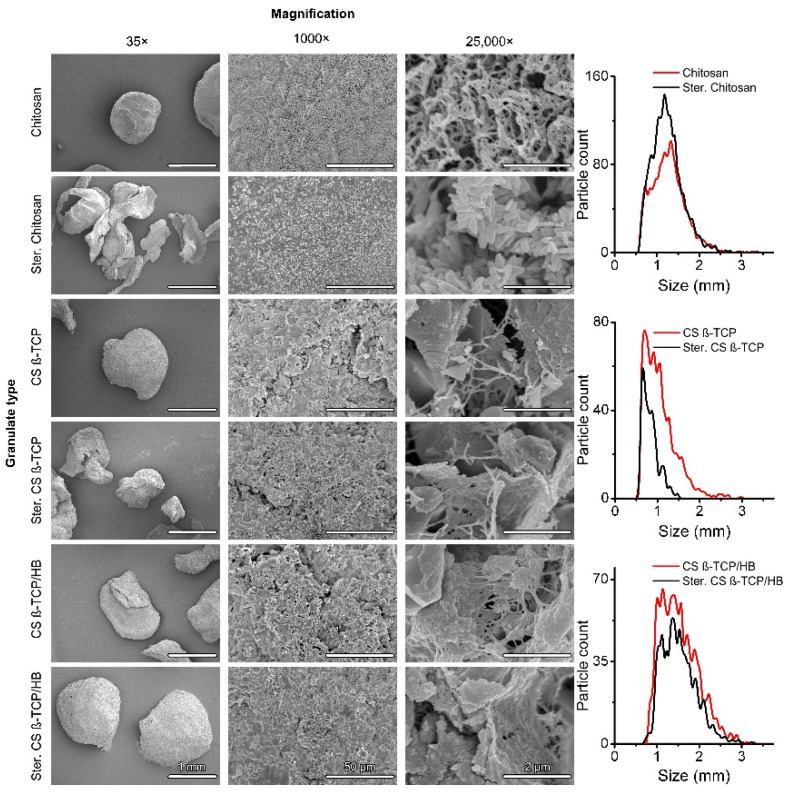
SEM images and size distributions of prepared granulates. The size of particles is understood as the diameter of a sphere, in which cross-section area equals the area of the object determined with digital image analysis.

**Figure 3 ijms-22-02324-f003:**
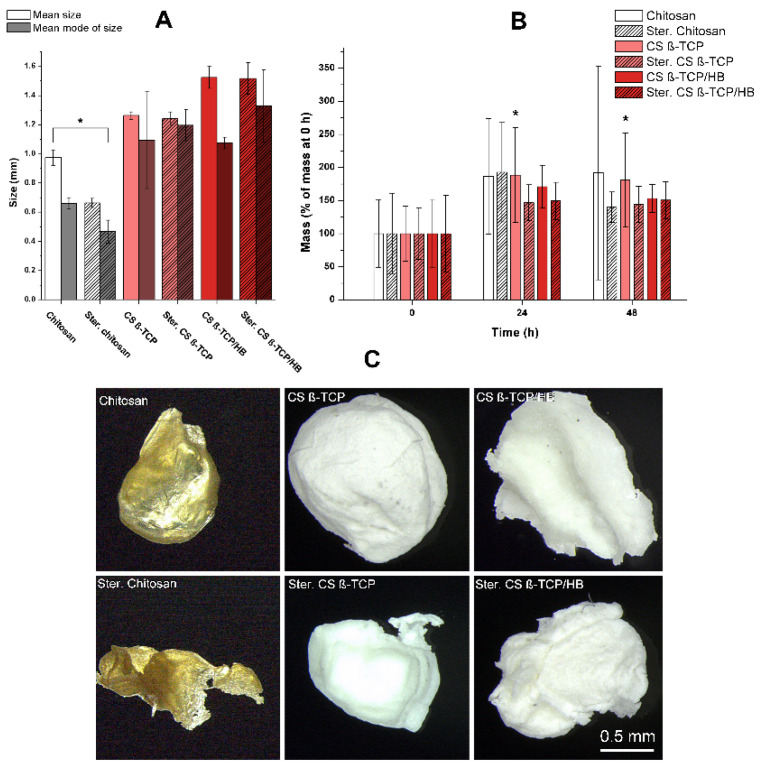
(**A**)—parameters of prepared materials (mean size, mean mode of size). The size parameter is understood as the cross-section of digitally photographed granules. Error bars represent standard deviations (*n* = 3). (**B**)—swelling of the materials after 24 and 48 h in 0.1 M PBS in 37 °C. Mean sizes marked with an asterisk (*) are significantly different from each other (one-way ANOVA with post hoc Tukey test, *p* < 0.05). Error bars represent standard deviations (*n* = 12). (**C**)—digital photography of materials after 48 h of swelling.

**Figure 4 ijms-22-02324-f004:**
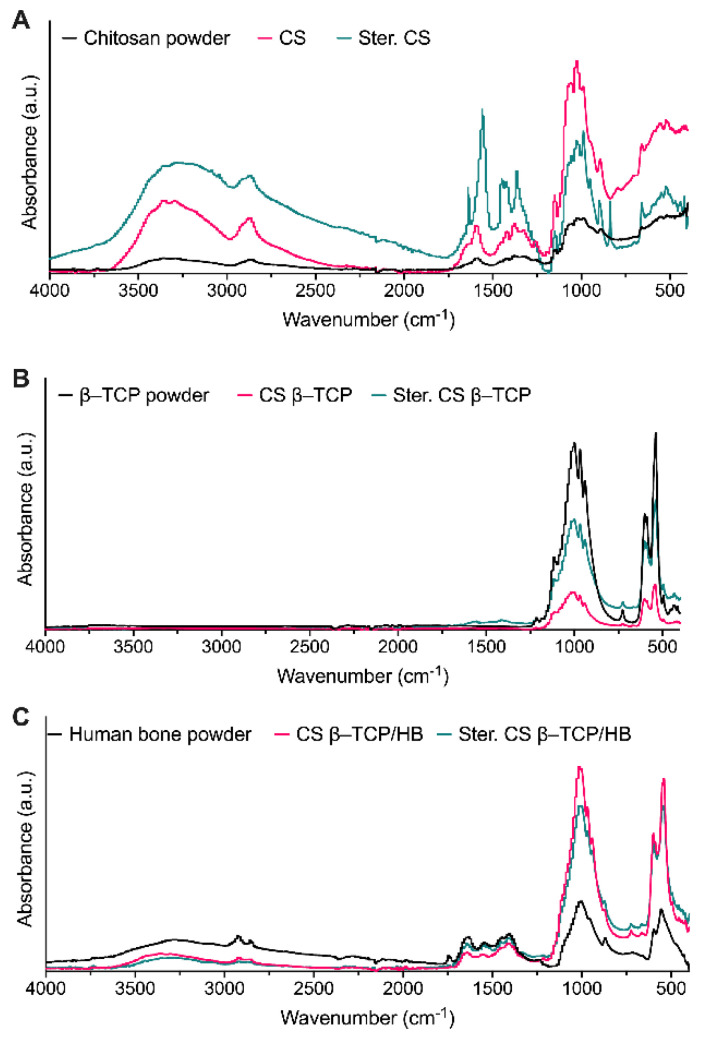
FTIR spectra of starting materials, granulates, and sterilized granulates: (**A**)—chitosan-based materials, (**B**)—materials with β-TCP only, and (**C**)—materials with β-TCP and human bone (HB).

**Figure 5 ijms-22-02324-f005:**
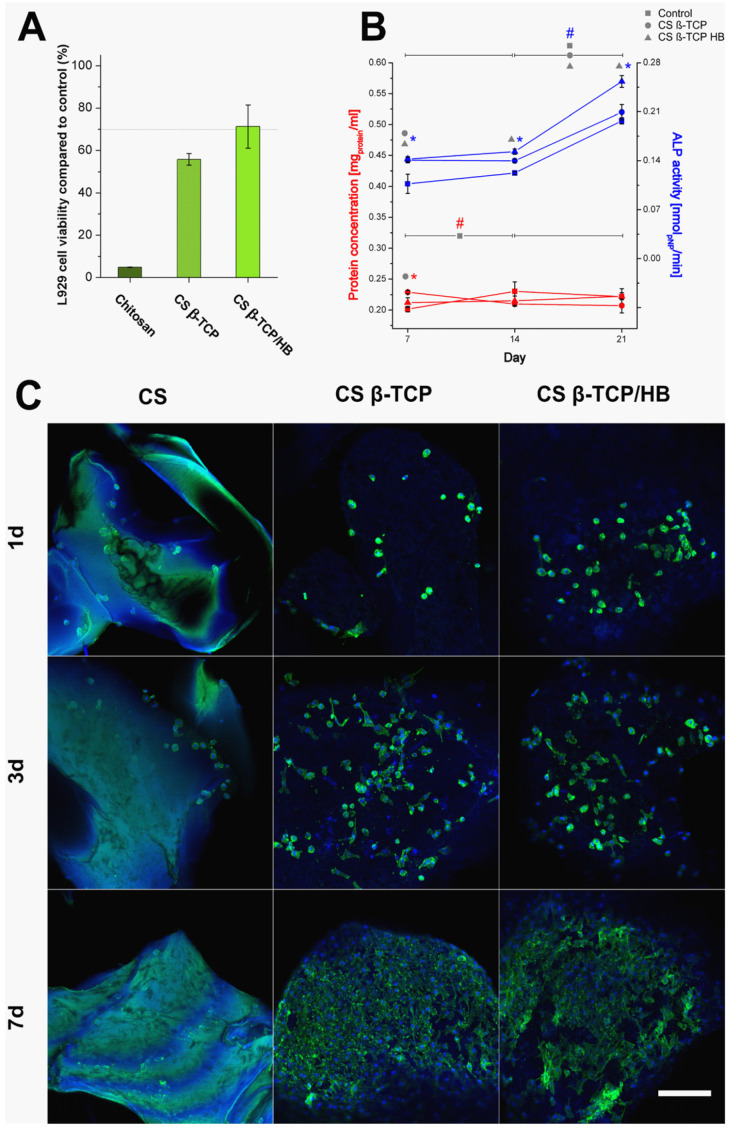
(**A**)—cell viability assay. L929 cells viability cultured for 24 h on extracts and determined with XTT assay on extracts (ISO EN 10993-5). Error bars represent standard deviation (*n* = 12). All means of viability are significantly different from each other (*p* < 0.05). (**B**)—alkaline phosphatase activity of control cells and granulates. ALP activity was measured in MG63 cell lysates, cells grown in 6-well plates with 20 mg of material per well. Total protein was determined with BCA assay. Asterisks (*) indicate significant differences (*p* < 0.05) between the marked sample mean values and control in the same week, while number signs (#) indicate significant differences (*p* < 0.05) between the marked sample mean values in time (weeks). (**C**)—confocal microscopy images of MG63 cell cultures on prepared materials. Nuclei are stained blue by DAPI, while the cytoskeleton is stained green by Alexa Fluor 488™ Phalloidin. Scale bar—200 µm.

## Data Availability

Data is contained within the Appendix A.

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
