# Peer review of "Chitosan-Human Bone Composite Granulates for Guided Bone Regeneration"

_ijms, 2021, doi:10.3390/ijms22052324_

Round 1
Reviewer 1 Report
1. I think you should check the overall paper with native speaker someone.
ex) Figure 1. Schematic of the CS b-TCP/HB composite material formation process---Figure 1. Schematic diagram for preparation of CS b-TCP/HB composites.
2. How did you obtain the particle size distribution as shown in Figure 2 ?
The distribution shape is very unusual..
3. Is the chitosan toxic? with aid of your data as shown in Fig. 5 (a), cell viability of chitosan is very lower than the others.
4. I am wandering why the protein concentrations were not changed with culture time as shown in Fig. 5 (b) compared to ALP activity was sharply increased with culture time from 14 to 21 days.
Author Response
Submitted changes have been highlighted in yellow .
- Thank you for the advice on the manuscript’s language. The document has been proof-read again and several linguistic corrections have been made. For convenience of reading linguistic corrections have not been highlighted.
- As has been described in Materials & Methods 3.2, page 10, line 315-322, the size of granules has been measured with the digital image processing method. During the measurement, the photography of the granulate is transformed into black & white bitmap, and the ImageJ software calculated the area of white particles (granules). Area value can be then used to calculate the sphere radius of a single particle. Neglecting area values smaller than a fixed value is necessary for the software to avoid counting single pixels or other small artifacts as granules. This explains the sudden 0.5 mm size cut-off on the distribution graphs. Appropriate description of the method and discussion of the results was added to the manuscript (132-134).
- As we have reported in Results section, cells cultured on pure, thermally sterilized chitosan particles did not pass the 70 % viability mark pointed in the ISO 10993:5 standard for cytotoxicity of biomaterials. We discussed the toxic effect of thermally sterilized chitosan on page 7, lines 210-216. We slightly changed this section to make it more explicit. However, the composite materials with calcium phosphate and bone were not toxic.
- The cell culture becomes stable after colonizing granulate particles. Then the cells start presenting their osteogenic activity and expressing ALP. Discussion on those results can be found on pages 7-8, lines 247-251.
Reviewer 2 Report
The manuscript topic is actual and the paper has merit. It could be attractive, adequate and interesting for the journal readers. However there are some point that authors should address in order to have a final more complete paper.
Authors should underline the limitation of the value of the study, and the clinical and surgical implication of the presented study should be added. At this stage the paper seems to be directed to researchers and not surgeons. Please emphasize the clinical application of the study, and its scientific rationale.
Remove the word "our" in all the sentences and replace it with "this" study, work, on so on....
The limitation of an "animal study or in vitro study" should be underlined and need to be synthesized in a paragraph.
....animal studies as well as in vitro study will only become more valid predictors of human reactions to exposures and treatments if there is substantial improvement in both their scientific methods as well as in more systematic review of the animal literature as it evolves. Systematic reviews of animal research or in vitro studies, if they are used to inform the design of clinical trials, particularly with respect to appropriate drug dose, timing and other crucial aspects of the drug regimen, will further improve the predictability of animal research in human clinical trials....
References are inadequate. Introduction section is poor. Some more references about the recent (2015-2020) CLINICAL reconstructive option just published have to be added.
At the same time discussion is poor.
In the discussion section authors should compare the results of the present study with others one presented and published in the literature.
Please add some samples as the following:
Other important marine bone substitutes material and clinical studies are the following, please add:
Cicciù, M.; Cervino, G.; Herford, A.S.; Famà, F.; Bramanti, E.; Fiorillo, L.; Lauritano, F.; Sambataro, S.; Troiano, G.; Laino, L. Facial Bone Reconstruction Using both Marine or Non-Marine Bone Substitutes: Evaluation of Current Outcomes in a Systematic Literature Review. Mar. Drugs 2018, 16, 27. https://doi.org/10.3390/md16010027
Author Response
Submitted changes have been highlighted in green .
- Additional introduction on clinical applications of granulates has been added (page 1, line 28-31).
- The possessives have been replaced with more neutral linguistic forms.
- The limitation of in vitro and further animal studies has been highlighted as requested, page 9, lines 278-282.
- References have been revised. Some of them, the ones which did not contribute to the point of the manuscript, have been removed, .
- Additional discussion has been added, especially in the topic of comparing obtained results to other literature findings.
Reviewer 3 Report
The manuscript deals with the preparation of novel biomaterials dedicated to bone tissue engineering. I recommend the publication after major revision including introduction and discussion section (e.g. FT-IR analysis) improvement as well as some additional physiochemical studies. The manuscript should contain XRD analysis as well as mechanical properties study. Also, the manuscript lacks swelling capability investigation.
Author Response
Thank you for your valuable insight with your suggestions of additional research. We have added the swelling results on all variants of the material, 24 and 48 hours in PBS, 37 °C. Results are added to the manuscript and properly described and discussed (Figure 3). Additional sections have been highlighted in cyan.
However, while we appreciate your suggestions, we would like to kindly question the necessity of XRD and mechanical studies. Obtaining XRD data in the laboratory we collaborate with demands a rather high amount of material and the measurement is destructive to the sample. Having a very limited supply of human bone approved for research we are not able to perform the experiment in satisfying and scientifically accurate quality. In addition, we have focused solely on the method of granulate production and its cytotoxicity according to ISO norm. Crystallographic data could be interesting but does not seem necessary to support the point we have presented in the paper. As for mechanical properties study, again it is a measurement that is destructive to the sample, but what is more, the granulate is not meant to be considered as load-bearing scaffold, which combines cell-adhesive properties and mechanical strength. The granulate is meant for filling bone defects, enhance regrowth of bone tissue and then facilitate further procedures, e.g., in dentistry prior implant insertion. Therefore, we would like to make an objection with the statement that mechanical studies of our material are essential for the point that this paper is trying to make. Nevertheless, we admit that our manuscript could describe our thesis and purpose of our material more precisely and we have rewritten the Introduction section. Submitted changes have been highlighted in cyan.
Round 2
Reviewer 2 Report
Paper is improved
Reviewer 3 Report
No further comments.